# Contribution of Meteor Flux in the Occurrence of Sporadic-E (Es) Layer over Arabian Peninsula

Muhammad Mubasshir Shaikh[1], Govardan Gopakumar[1,3], Aisha Abdulla Alowais[2], Maryam Essa Sharif[2], Ilias Fernini[1,2,3]

[1]Space Weather and Ionosphere Laboratory, Sharjah Academy for Astronomy, Space Sciences and Technology, University of Sharjah, Sharjah, UAE.
[2]UAE Meteor Monitoring Network, Sharjah Academy for Astronomy, Space Sciences and Technology, University of Sharjah, Sharjah, UAE.
[3]Department of Applied Physics & Astronomy, University of Sharjah, Sharjah, UAE.

*Correspondence to*: Muhammad Mubasshir Shaikh (mshaikh@sharjah.ac.ae)

**Abstract:** Sporadic-E (Es) layer is generally associated with a thin-layered structure present in the lower ionosphere, mostly consisting of metallic ions. This metallic ion layer is formed when meteors burn in the upper atmosphere resulting in the deposition of free metal atoms and ions. Many studies have attributed to the presence of the Es layer due to the metallic ion layer, specifically when the layer is observed during the nighttime. Using data from a network of meteor monitoring towers and a collocated digital ionosonde radar near Arabian Peninsula, in this paper, we are reporting our observations of Es layer occurrences together with the meteor count. The trend of monthly averages of Es layer intensity shows a maximum in late spring and early summer months and a minimum in winter months whereas the meteor counts were highest in winter months and lowest in spring and early summer months. This shows that the presence of the Es layer and the meteor counts have no correlation in time, both diurnally and seasonally. This leads us to conclude that the presence of meteors is not the main cause of presence of the Es layer over Arabian Peninsula.

Key words: Sporadic-E, Meteor Flux, foEs, Ionosphere

## 1 Introduction

Meteors are the visible appearance of extraterrestrial dust, generally known as meteoroids. They appear in the sky when meteoroids ablate in the Earth's atmosphere. Meteors can be categorized as being either part of a shower or of the background meteor flux. There are a vast amount and variety of meteoroid material entering the atmosphere every day (Ceplecha et al., 1998), and its deposition is highly variable spatially as well as temporally. These variations are attributed to the inconsistency of the meteoroid material density surrounding the Earth, seasonal changes of atmosphere and the Earth's movement around the Sun, the methods of observing them such as the geographical location of the observing site and geometrical factors related to the observing instruments' capability and position of sources etc. This extraterrestrial influx changes the metallic composition of the Earth's atmosphere and lower ionosphere. This happens when meteors burn in the dense atmosphere, resulting in the heating and deposition of free metal atoms and ions (Ceplecha et al., 1998). It is now a well-established fact that, permanent ionized metal layer in lower ionosphere, at around 90-130 km altitude, is due to the ablation of meteors in that region (Plane et al., 2015). Meteor observations can be performed with the radio (Stober and Chau, 2015; Lima et al., 2015; Yi et al., 2016) as well as with visual means (Vitek and Nasyrova, 2018; Kozlowski et al. 2019; Fernini et al., 2020). Detection using visual cameras can only be performed during the night compared to radio-based observations that can be performed throughout the day and suitable for estimating total meteor activity. A combination of multiple types of observations may also be used (Brown et al., 2017).

Kopp (1997) showed that the thin-layered structured sporadic-E (Es) layer in the Earth's ionosphere, lying between the altitude range of 90-130 km, is mostly consisted of ionized metal atoms $FeC$, $MgC$ and $NaC$. In mid-latitudes, the so-called 'windshear' theory is thought to be the mechanism responsible for this formation (Whitehead, 1989). Therefore, the intensity and occurrence of the Es layer is expected to be proportional to the amount of metal ion content at the lower ionosphere and its chemical processes, as well as meteorological processes in the lower ionosphere (Feng et al., 2013; Yu et al., 2015). The nature of the Es layer observed globally has been a function of many factors such as geographical latitude, observing instruments' sensitivity of the viewing system etc. For example, the Es layer can be observed at almost all times at some geographical locations around the globe (Shaikh et al. 2020); thus, making the term 'sporadic', misleading. The behavior of the Es layer over the Arabian Peninsula has not been studied by many. Recently, Shaikh et al. (2020a; 2020b) demonstrated the relationship between L-band

scintillation and the occurrence of the Es layer over the Arabian Peninsula. The study also revealed a consistent presence of the Es layer during the nighttime hours, between sunset and sunrise.

In this paper, we are reporting the observations of the Es layer and the meteor counts simultaneously observed during nighttime over the Arabian Peninsula region for the first time. A well-established presence of the Es layer can be observed during all daytime and nighttime hours with higher intensity around midday hours and lesser intensity at early morning and nighttime hours. A consistent meteor count is also present throughout the 1-year observation period (May 2019 – April 2020), reported in this work. It has been observed that presence of meteors is not the main cause of the presence of nighttime Es over Arabian Peninsula since the Es layer intensity (average critical frequencies of the Es layers (foEs)) show no seasonal correlation with the number of meteors observed. The dependence of Es layer intensity (foEs) due to meteor count has been calculated using linear correlation coefficients. Negative values of correlation coefficient show an anti-correlation relationship between the two data sets.

## 2 Data and Methodology

The meteor counts for this study has been obtained in collaboration with the UAE Meteor Monitoring Network (UAEMMN) project (Fernini et al., 2020). The project aims to monitor and detect meteor occurrences in the region above the United Arab Emirates from sunset to sunrise. To achieve this, three monitoring towers have been constructed and installed in different parts of the country. For each tower, sixteen cameras are distributed along with a ring-like structure with lenses of 6mm and 8mm, while the 17th camera utilizes a wide-angle lens and is located at the center of the structure (Fernini et al., 2020). Following a simulation using Systems Tool Kit software (STK: https://www.agi.com/products/50stk) as shown in Fig 1a, the towers' locations were selected as illustrated in Fig 1b (made using © Google Maps). In Fig 1, the green color represents the area of the sky covered by the 8mm lenses, while the red represents the coverage of the 6mm lenses. The yellow squares show what the wide-angle lens can see and cover. Thus, the STK simulation illustrates how much each tower covers the UAE sky, which adds up to 70% coverage of the sky. Each of the three UAEMMN towers employs the use of the UFOCapture Software developed by SonotaCo (SonotaCo, 2005) to detect meteor occurrences. The software can detect movements from the feed of the cameras on the towers. If a movement or action is detected, it writes the video of the action to the hard disk of the computer, from a few seconds before the action is recognized to a few seconds after the action is completed. During the night, the bright streaks produced by a meteor burning up in the atmosphere allows the software to detect movements from the sudden changes easily in pixel values.

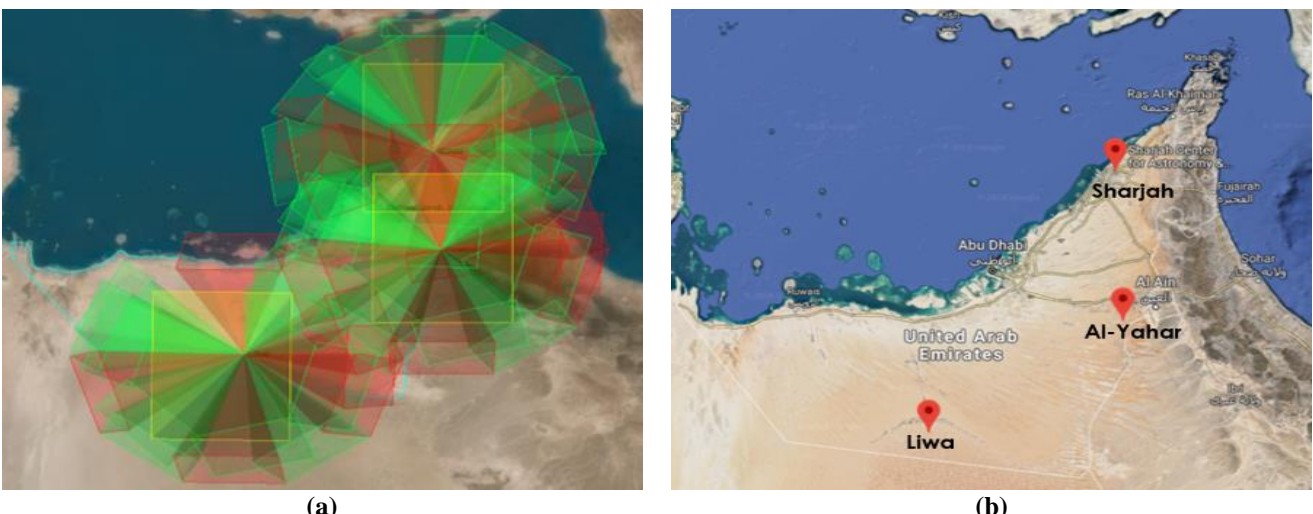

(a)          (b)

**Figure 1:** (a) Sky coverage simulation by all cameras using Systems Tool Kit (STK). (b) Location of the towers pinpointed on the UAE map using © Google Maps.

Two other software, UFOAnalyzer and UFOOrbit, also developed by SonotaCo (SonotaCo, 2007a; SonotaCo, 2007b), are used to calculate parameters that define the meteorite. UFOAnalyzer can calculate the direction and elevation of the meteorite occurrence. If the meteorite is detected by two or more sites, UFOOrbit can calculate the orbit and the radiant point of the meteorite. Fig 2 shows a radiant map obtained as the result of analyses by the software. The radiant map shows radiant points on a sinusoidal projection map of the observed meteors, which **is** defined as the point in the sky from which the path of the observed meteor begins. For a radiant point to be plotted on the map by the software, a double detection of the meteor should occur, meaning that two cameras from at least two different towers need to observe the same meteor. Fig 2 shows the radiant

points of meteors observed by the Sharjah and Al-Yahar towers during the period between May 2019 and April 2020. On the map, constellations such Orionids and Taurids are denoted as J5_Orio, J5_nTa and sTa, respectively. Hence, the radiant points that are close to a constellation imply that they belong to the respective meteor group. In this figure, there are meteors that belong to the Orionids meteor shower, as well as Southern and Northern Taurids and several others, in addition to sporadic meteors that do not belong to any shower. By locating the radiant maps, the network ensures its accuracy in terms of linking a meteor to its respective shower. The radiant velocity is color coded as shown in the figure.

| Instruments | Geographical Lat. | Geographical Long. | Specification |
|---|---|---|---|
| Sharjah Digital Ionosonde | 25.285381ºN | 55.464417ºE | Freq. Range = 1-30 MHz |
| Sharjah Meteor Monitoring Tower | 25.235611ºN | 55.539645ºE | CCD Cameras |
| Al-Yahar Meteor Monitoring Tower | 24.285922ºN | 55.463625ºE | CCD Cameras |
| Liwa Meteor Monitoring Tower | 23.104722ºN | 53.754828ºE | CCD Cameras |

**Table 1:** Location of the instruments used to generate data for this study

The critical frequency of the Es layer (foEs) of the ionosphere is obtained from the ionosonde collocated with the Sharjah meteor monitoring tower. The ionosonde records one ionogram every 15 minutes, and it has been in operation since May 2019. All ionogram-derived parameters used in this study have been manually scaled. All the data used in this study are available from SWI Lab, (2020). Since the data from the meteor towers is only available from nighttime observations and the data from the ionosonde is observed throughout the day and night, the daily Es intensity (average foEs value) has been used to compare with the daily meteor count to study the impact of the number of meteors present and their influence on the presence of Es (Haldoupis et al., 2007).

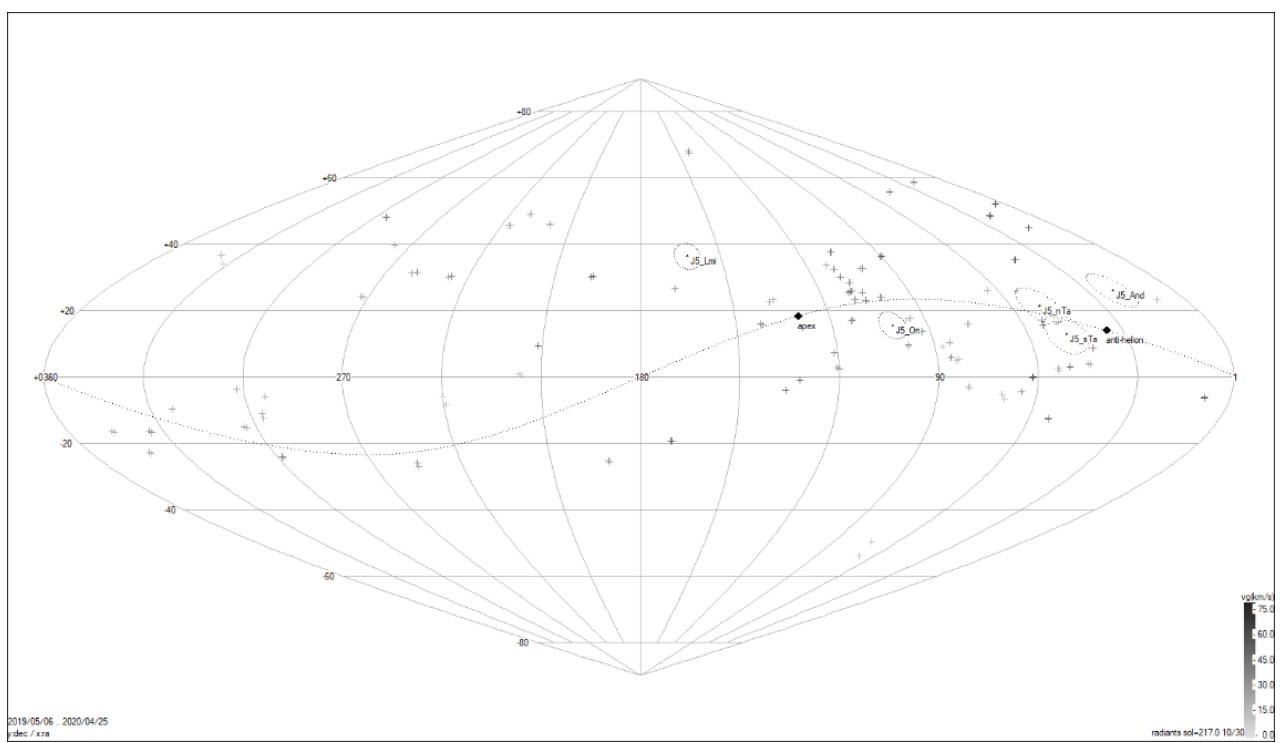

**Figure 2**: A radiant map of meteor observations by the Sharjah and Al-Yahar stations during the period May 2019 – April 2020

**3 Discussion**

Fig 3 shows the observation of the Es layer and meteor count. Fig 3(a) and 3(b) show that a constant presence of Es can be observed throughout the year and all hours of the day with higher intensity (average foEs) around midday hours and lesser intensity at early morning and nighttime hours. An important point to note here is that this observation was performed during a time when the solar activity was low. The average F10.7 solar radio flux value during a 1-year observational period was

recorded as 69.43 sfu. Only geomagnetically quiet days with an average daily Kp value less than 3 were selected for the analysis. It is expected that the Es layer observations would be more substantial as the solar cycle 25 gets stronger in coming years. Fig 3(c) shows the hourly meteor count for the whole 1-year observational period. No observations were recorded during
the daytime.

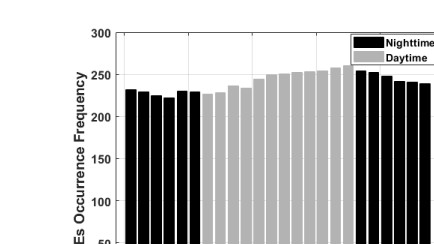 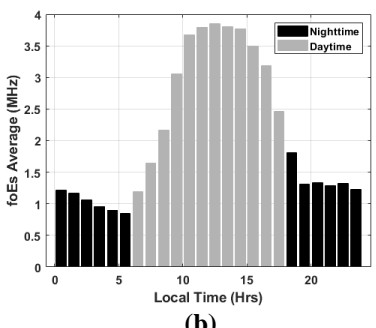 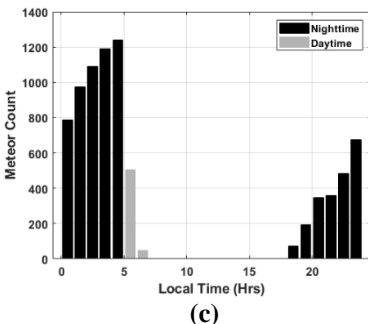

(a) (b) (c)

**Figure 3:** Simultaneous monitoring of meteors and the Es layer over Arabian Peninsula from May 2019 – April 2020. (a) Es occurrence frequency as function of local time, (b) Hourly average of foEs recorded using ionosonde, (b) Hourly meteor count

Fig 4 shows a comparison between the daily and monthly meteor counts with daily and monthly averages of foEs. 4(a) shows
all daily observations (24 hours), and 4(b) provides observations for nighttime only. The trend of monthly averages of the Es layer intensity shows a maximum in late spring and early summer months and a minimum in winter months (except for a slight peak in January). At the other end, the monthly meteor count shows an opposite trend with a larger number of meteors observed during November - December 2019 and very low numbers in the Spring and Summer months. Both 4(a) and 4(b) shows a very similar trend for foEs averages. The difference is in the intensity of the Es layer which is greater when all observations are
considered due to the inclusion of daytime the Es layer observations. The meteor count is the same in both cases since we have only observed meteors through visual cameras during the nighttime.

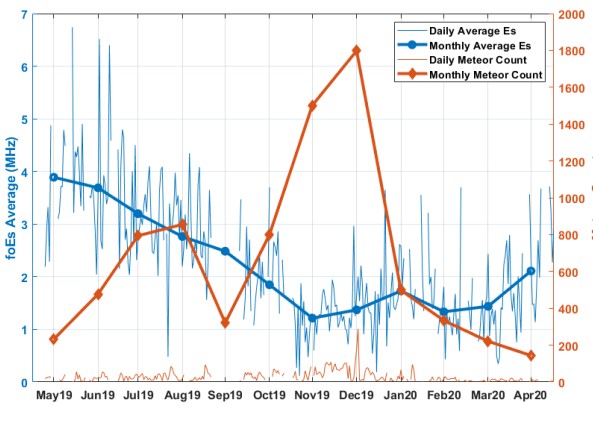 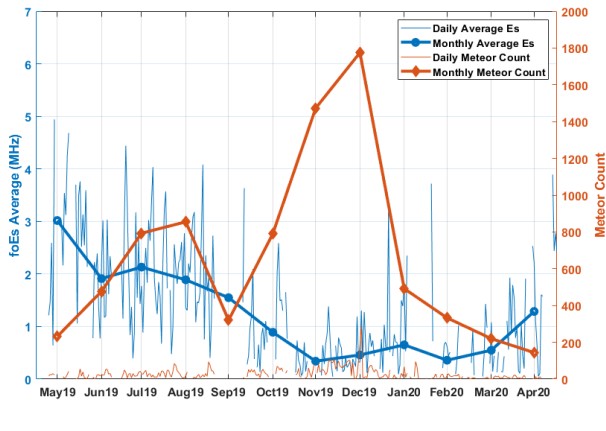

(a) (b)

**Figure 4:** Daily and monthly averages of foEs and meteor count over Sharjah. (a) Including all observations (24 hours), (b) Nightime observations only

130 The observations presented in Fig 4 are inconsistent with Younger, et al. (2009) who reported meteor flux data observed by radars installed at Esrange (68ºN), Ascension Island (8ºS), and Rothera (68ºS). They showed that, for high latitudes, there is a clear annual cycle present where the maximum count rate is observed in summer. Whereas for low latitude Ascension Island, the maximum count rates were observed for both solstices (summer and winter). Similar observations were also reported by Singer et al. (2004) using a meteor radar situated at the ALOMAR observatory (69ºN), and Haldoupis et al. (2007) from
135 European latitudes.

There have been other studies that correlate meteor activity with the Es layer seen in ionograms, examples of which include Chandra et al. (2001), Haldoupis et al. (2007), and Ellyet and Goldsborough (1976). There are also numerous studies whose results are inconclusive. For example, Baggaley and Steel (1984) were unable to find any correlation between meteor activity
and the Es layers' occurrence. Kotadia and Jani (1967) reported that they did not find any increase in the occurrence of the Es

layers during a period of anomalously large increase in meteor incidence in 1963, but instead found that the Es layers were formed less frequently during that period, suggesting an inverse relationship between the formation of the Es layers or meteor incidents. The results presented in this paper also follow a similar pattern, with foEs decreasing significantly during the period between October 2019 to January 2020; even with the increased meteor count during that period (see Fig. 4). This may be because plasma density abnormalities may exist which may cause ionograms to record scatter echoes beyond the foEs. Cross-field plasma instabilities cause the abnormalities due to the various electrodynamic processes in the ionosphere. These instabilities are triggered by the enhancement of plasma density in a particular volume when an external force acts on that same volume. A small disturbance can then lead to the separation of charges, which produces a small electric field, which with the presence of the geomagnetic field increases the disturbance (Simon, 1963). Meteoric activity may provide metallic ions to the ionosphere, but they may not be displayed in ionograms if the conditions are unfavorable. The aforementioned instabilities have been shown to be capable of producing the diffuse type of the Es layer (Tsuda et al., 1969). The formation of this diffuse layer may cause the ionogram to display scatter echoes that exceed the actual critical frequency of the sporadic E layer formed as a result of metallic ions deposited by meteors. This may be why a good correlation between meteor activity and the Es layer is not seen (Chandra et al., 2001) and, also confirmed by the correlation plot in Fig 5. It is shown in Fig 5 that the annual variation of both observations, on average, does not correlate monthly, having linear correlation coefficients less than -0.35 (negative 0.35) for both full-day and nighttime observations.

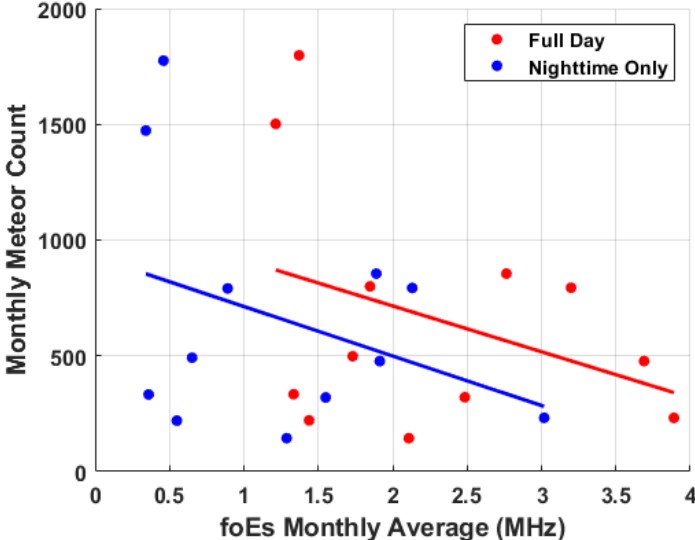

**Figure 5:** Relationship betweeen foEs monthly averages and monthly meteor count observed at Sharjah.

Fig 4 shows differences between the variations in foEs and meteor counts observed both at small and large timescales. The Es layer may be affected by differences in climatology and wind dynamics. For example, long-period zonal and meridional winds at the mesopause region, with periods between 2 to 18 days may be considered to be planetary wave activity. Planetary waves have been observed to have strong variability between different seasons, with periods of 2 days in the summer, 5 days in spring and even exceeding 10 days during the winter (Jacobi et al., 1998). Studies have proposed vortex flows associated with planetary waves to explain the seasonal dependance of sporadic E-layers (Shalimov et al., 1999). Vortex flows are already known to affect the development of E-layers (Pancheva et al. 2003). The meteor count may also be influenced by some biases. A number of the recorded meteors may not be metallic in nature and would not deposit any metallic ions in the ionosphere, possibly explaining why a higher meteor count during winter months did not amount to a higher average foEs. Nevertheless, visual meteor counts may not include all meteors. The metallic ions deposited by a meteor in the ionosphere may not be proportional to the meteoric activity as well (Haldoupis 2007). The exact relationship between metallic ion densities and meteoric activity is unknown, and the transportation of metallic ions by neutral winds is not accounted for. Due to these uncertainties, the incongruous relationship between foEs and visual meteors count is not unexpected, however, they are not enough to explain the incongruity. Another possible scenario arises when neutral winds are considered, which could transport metallic ions to the local ionosphere in study irrespective of the number of observed meteors (Haldoupis 2007). This may be an explanation of the inverse correlation between foEs and meteor counts observed during summer months.

| Constellation | Hourly Dates | Rate | Speed (km/s) | Shower Name | Quantity from the UAEMMN towers |
|---|---|---|---|---|---|
| Capricorn | Jul 3 – Aug 15 | 5 | 41 | Capricornids | 6 |
| Perseus | Aug 10 - Aug14 | 40 | 60 | Perseids | 2 |
| Taurus | Nov 01 – Nov 07 | 8 | 30 | Taurids | 10 |
| Gemini | Dec 10 – Dec 13 | 50 | 35 | Geminids | 17 |
| Monocerous | Dec 5 – Dec 20 | 15 | 35 | Monocerotids | 2 |
| Hydra | Dec 03 – Dec 15 | 3 | 58 | Hydrids | 4 |

**Table 2:** Meteor showers observed by UAEMMN network

One can expect to see a meteor entering in the Earth's atmosphere every 10 minutes or so, but there are predictable times during the year when the Earth's atmosphere is full of them, and these are referred to as meteor showers (Kronk, 2014). These showers occur monthly with some meteor showers more pronounced than others, depending on their parents' progenitors (Collins, 2020). We can see about 30 meteor showers during the year. Since the meteors in each shower seem to come from a certain point in the sky, the shower is named after the constellation from which the meteors come. The Quadrantids, the Perseids, and the Geminids are the most prominent of all the meteor showers. Table 2 is showing the data obtained from the UAEMMN network about the meteor showers. The data is taken from the same one-year study period used in this work. We can clearly observe that most meteor showers occurred from the period from August to December resulted in significant increase in the numbers of visual meteors observed in UAE (see Fig 4). However, it seems quite understandable here that not all those meteor showers contributed to the presence of the Es layer in UAE since the Es layer observations were higher in summer than during the winter months.

The Es layer may not be observed if the meteoric activity period does not provide long-lived metallic ions in the background plasma density. However, under favorable conditions, the meteoric debris consisting mostly of metallic ions could be converged to form sharp layers of ionization leading to density gradients responsible for ionospheric irregularities and spreading of the echos in the ionograms. Since the ionospheric background conditions considerably vary with latitudinal region, simultaneous observations from different geographical regions would be needed to confirm a certain meteoric activity and its linkage with the appearance of the Es layer. Therefore, a thorough analysis using the systematic analysis of past data taken simultaneously from different latitudinal regions yield a better picture of the role of meteoric activity in the E-region ionization.

## 4 Conclusion

In this paper, simultaneous observations of foEs and the meteoric influx (meteor count rates through visual cameras) show no diurnal or seasonal dependence over Arabian Peninsula. We report the seasonal observations of the Es layer simultaneously taken with the visual count observations from a geographical region which has not been reported before. However, no attempt was made to link the simultaneous observation of the Es layer and meteor influx in detail.

Our one-year observations clearly show that the Es layer intensity is not dependent on the presence of meteor flux since the meteor count trend, which is peaking in winter and declining in summer, is found to be uncorrelated to the trend observed for Es layer intensity (see Fig 4 and Fig 5). This may have happened because plasma density abnormalities may exist which may cause ionograms to record scatter echoes beyond the foEs. The abnormalities are caused by plasma instabilities due to the various electrodynamic processes in the ionosphere. Meteoric activity may provide metallic ions to the ionosphere, but they may not be displayed in ionograms if the conditions are unfavorable. This may have been the reason why a good correlation between meteor activity and the Es layer intensity cannot be seen by our two collocated instruments. Such results have been rarely reported in the literature and do not comply with frequently reported studies which established a strong seasonal correlation between daily meteor counts with daily averages of the Es layer occurrences, as mentioned in the references above. It is also important to note that this study, unlike many of the previous studies, used visual observations for observing meteors. Since the data is manually checked and verified from the recorded visual data, unlike for radio based radar observations where rate of false observations is very high, the study is likely to provide a real picture since there is a very little chance of having false data. Nevertheless, the authors believe that a more detailed study is required to fully investigate and properly identify the Es layer seasonal dependence on the meteor influx in the region around Arabian Peninsula.

**Data Availability**

All data used in this work is available from the dataverse of SWI Lab and acquired and managed by Sharjah Academy for Astronomy, Space Sciences and Technology (see reference SWI Lab, (2020)).

**Author Contribution**

Muhammad Mubasshir Shaikh: Conceptualization, Investigation, Data Curation, Writing – Original Draft.
Govardan Gopakumar: Investigation, Software, Data Curation, Writing - Review & Editing.
Aisha Abdulla Alowais: Software, Writing - Review & Editing
Maryam Essa Sharif: Software Writing - Review & Editing
Ilias Fernini: Writing - Review & Editing

**Competing Interests**

The authors declare that they have no conflict of interest.

**Acknowledgement**

Authors are thankful for the two anonymous reviewers for their valuable comments which helped improve the quality of the paper.

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
