# Peer review of "Contribution of Meteor Flux in the Occurrence of Sporadic-E (Es) Layer over Arabian Peninsula"

_Annales Geophysicae, 2020_

## Referee Comment (RC1) · Anonymous Referee #1 · 17 Dec 2020

This paper is very interesting, but it is necessary significant improvements before publication. Please, consider the suggestions following:

1. The main point is the lack of manuscript organization. The authors need to include more information and the discussion about the anti-correlation. 2. Abstract- Lines 13-15: The authors need to include more details about the results. 3. Introduction: I suggest that authors include 2 or 3 paragraphs about the previous study of the Es layer behavior over the Arabian Peninsula and the wind shear and meter relationship. 4. The last paragraph of the Introduction (lines 43-45) needs to be more specific with the authors' results in this work. 5. The authors need to explain Figure 1 and Figure 2 in more detail and maybe include the examples of ionograms since the Es layer are different forms in each region. 6. Lines: 109-110: Haldoupis et al. (2007) only talk that

the Es layers and the meteoric influx follow a similar seasonal dependence marked by a strong summer maximum only. The phrase seems that the results of Haldoupis et al. (2007) are related with both solstices. 7. The authors mentions that the "There are also numerous studies whose results are inconclusive", but they do not perform a deeply discussion about to works.

8. Figure 5 should be in discussions instead of conclusions. 9. Lines 157-159: The authors could give more proposals because this behavior occurs over the analyzed region. 10. Do the authors only consider the nighttime values of foEs to perform the correlation in Figure 5? Does this anti-correlation in this Figure not occur because the Es layer is very dynamic? 11. There are many mistakes in the language (for example, relationship in Figure 5), and the English need to be improved. 12. Please, consider including the word layer after the Es.

---

## Referee Comment (RC2) · Anonymous Referee #2 · 22 Dec 2020

The paper presents a topic of sure interest that can stimulate the curiosity of a number of scientists because it poses questions still unsolved and because the analysis is based on measurements taken in a region scarcely reported in the open literature. This is the reason why I would be in favour of the paper publication. Nevertheless, my major concern is about the Es definition: are the authors considering the foEs or the appearance of Es? I see some confusion in the manuscript between the two quantities and I wonder why the authors are accounting for foEs instead of focusing on the occurrence of Es. Here follow some major issues that the authors might consider. In the Discussions: You write "A constant presence of Es can be observed throughout the year and all hours of the day", but the persistence throughout the year is not visible from your plot in which you identify the hourly average foEs as function of the day

time. To show such consistency you should include the count. You write "The trend of monthly averages of Es clearly shows a rise in summer months and a decline in the autumn and winter months." From the plot I can see an increase in springtime and a decrease from early summer. In figure 5 are you reporting the foEs or the count of Es occurrence?

Minor revisions Fig.3 caption: meteor instead of metero Lines 106-110 page 4: replace o with $^\circ$ Line 119 page 4: Fig 4 shows

---

## Author Comment (AC1) · 3 Jan 2021

Reviewer: This paper is very interesting, but it is necessary significant improvements before publication. Please, consider the suggestions following:

Authors' Response: Authors would like to thank reviewer for spending his/her precious time to read and comment on the paper. The comments from the reviewer have certainly helped improve the quality of the paper. All changes made with reference to the reviewers' comments are highlighted in red in the revised manuscript.

Reviewer: 1. The main point is the lack of manuscript organization. The authors need to include more information and the discussion about the anti-correlation.

Authors' Response: Considering reviewer's comment, following sentences have been

added at the end of "Introduction" section:

"Correlation coefficient in this paper has been used to study the linear relationship of meteor counts and the intensity of Es layer in their diurnal and seasonal variation. On the contrary, anti-correlation would mean that variations of the two independent observations during the observational period have opposite linear trends."

Reviewer: 2. Abstract- Lines 13-15: The authors need to include more details about the results.

Authors' Response: Last sentence of the Abstract has been modified as follows to indicate more details about the outcomes of the study:

"The trend of monthly averages of Es layer intensity shows a maximum in late spring and early summer months and a minimum in winter months whereas the meteor counts were highest in winter months and lowest in spring and early summer months. This shows that the presence of Es layer and the meteor counts have no correlation in time, both diurnally and seasonally, which leads us to conclude that the presence of meteors is not the main cause of the presence of Es layer over Arabian Peninsula."

Reviewer: 3. Introduction: I suggest that authors include 2 or 3 paragraphs about the previous study of the Es layer behavior over the Arabian Peninsula and the wind shear and meter relationship.

Authors' Response: The authors were not able to find any previous studies about the relationship between windshear and meteors over the Arabian Peninsula. The only study reporting the observations of meteors over Arabian Peninsula is by Fernini et al., (2020)." Following para has been added for the previous studies found which studied the behavior of Es Layer:

"The behavior of Es layer over Arabian Peninsula has not been studied by many. Recently, Shaikh et al. (2020a; 2020b) demonstrated the relationship between L-band scintillation and the occurrence of the Es layer over the Arabian Peninsula. The study
also revealed a consistent presence of Es layer during the nighttime hours, between sunset and sunrise."

Shaikh, M., Gopakumar, G., Hussein, A., Kashcheyev, A., & Fernini, I. (2020b). Daytime GNSS scintillation due to Es over Arabian Peninsula during low solar activity. Results In Physics, 20, 103761. doi: 10.1016/j.rinp.2020.103761.

Reviewer: 4. The last paragraph of the Introduction (lines 43-45) needs to be more specific with the authors' results in this work.

Authors' Response: Following line have been added at the end of the 'Introduction' section:

"It has been observed that the presence of meteors is not the main cause of the presence of nighttime Es over Arabian Peninsula since the Es layer intensity (average critical frequencies of the Es layers (foEs)) show no seasonal correlation with the number of meteors observed."

Reviewer: 5. The authors need to explain Figure 1 and Figure 2 in more detail and maybe include the examples of ionograms since the Es layer are different forms in each region.

Authors' Response: Considering reviewer's comment, the first paragraph of the section 'Data and Methodology' has been modified to include more details about Figure 1, as follows:

"The meteor counts for this study has been obtained in collaboration with the UAE Meteor Monitoring Network (UAEMMN) project (Fernini et al., 2020). The project aims to monitor and detect meteors' occurrences in the region above the United Arab Emirates from sunset to sunrise. To do so, three monitoring towers have been constructed and installed in different parts of the country. For each tower, sixteen cameras are distributed along a ring-like structure with lenses of 6mm and 8mm, while the 17th camera utilizes a wide-angle lens and is located at the center of the structure

[Figure]

(Fernini et al., 2020). Following a simulation using Systems Tool Kit software (STK: https://www.agi.com/products/50 stk) as shown in Fig 1a, the towers' locations were selected as illustrated in Fig 1b (made using © Google Maps). In Fig 1, green color represents the area of the sky covered by the 8mm lenses, while the red represents the coverage of the 6mm lenses. The yellow squares show what the wide-angle lens can see and cover. Thus, the STK simulation illustrates how much each tower covers of the UAE sky, and this adds up to 70% coverage of the sky. Each of the three UAEMMN towers employs the use of the UFOCapture Software developed by SonotaCo (SonotaCo, 2005) to detect meteor occurrences. The software can detect movements from the feed of the cameras on the towers. If a movement or action is detected, it writes the video of the action to the hard disk of the computer, from a few seconds before the action is recognized to a few seconds after the action is completed. During the night, the bright streaks produced by a meteor burning up in the atmosphere allows the software to easily detect movements from the sudden changes in pixel values."

The second paragraph of the section 'Data and Methodology' has been modified to include more details about Figure 2, as follows:

"Two other software, UFOAnalyzer and UFOOrbit, also developed by SonotaCo (SonotaCo, 2007a; SonotaCo, 2007b), are used to calculate parameters that define the meteorite. UFOAnalyzer can calculate the direction and elevation of the meteorite occurrence. If the meteorite is detected by two or more sites, UFOOrbit can calculate the orbit and the radiant point of the meteorite. Fig 2 shows a radiant map obtained as the result of analyses by the software. The radiant map shows radiant points on a sinusoidal projection map of the observed meteors, which is defined as the point in the sky from which the path of the observed meteor begins. For a radiant point to be plotted on the map by the software, a double detection of the meteor should occur, meaning that two cameras from at least two different towers need to observe the same meteor. Fig 2 shows the radiant points of meteors observed by the Sharjah and Al-Yahar towers during the period between May 2019 and April 2020. On the map, constellations such

Orionids and Taurids are denoted as J5_Orio, J5_nTa and sTa, respectively. Hence, the radiant points that are close to a constellation imply that they belong to the respective meteor group. In this figure, there are meteors that belong to the Orionids meteor shower, as well as Southern and Northern Taurids and several others, in addition to sporadic meteors that do not belong to any shower. By locating the radiant maps, the network ensures its accuracy in terms of linking a meteor to its respective shower. The radiant velocity is color coded as shown in the figure."

Following figure (attached as Fig. 1) shows selected ionograms obtained during a 24-hour period from LT 4:00 (UT 0:00) on July 31, 2019 to LT 4:00 on August 1, 2019. It can be observed that Es layer traces are present persistently over the 24-hour period. Most of the traces indicate an f-type (flat) Es layer, however, c-type Es layers are also seen infrequently.

Reviewer: 6. Lines: 109-110: Haldoupis et al. (2007) only talk that the Es layers and the meteoric influx follow a similar seasonal dependence marked by a strong summer maximum only. The phrase seems that the results of Haldoupis et al. (2007) are related with both solstices.

Authors' Response: In the para mentioned by the reviewer (Lines: 109-110), authors have mentioned that there are numerous studies showing that the meteor count rate peaks in summer which was not observed by the data collected for this study. Haldoupis et al. (2007) showed the same by using radio frequency radar data from the mid-latitudes. In addition, in Fig. 2 of their article, Haldoupis et al. (2007) clearly showed that there is also a good correlation present between mean daily foEs and meteor counts throughout the year; in all four seasons.

Reviewer: 7. The authors mentions that the "There are also numerous studies whose results are inconclusive", but they do not perform a deeply discussion about to works.

Authors' Response: Authors are thankful for the reviewer for pointing out the missing information. Following lines has been added in the manuscript for references:

"For example, Baggaley and Steel (1984) were not able to find any correlation between meteor activity and occurrence of Es layers. Kotadia and Jani (1967) reported that they did not find any increase in the occurrence of Es layers during a period of anomalously large increase in meteor incidence in 1963, but instead found that Es layers were formed less frequently during that period, suggesting an inverse relationship between the formation of Es layers or meteor incidents. The results presented in this paper also follow a similar pattern, with foEs decreasing significantly during the period between October 2019 to January 2020; even with the increased meteor count during that period (see Fig. 4)."

Baggaley, W. J., & Steel, D. I. (1984). The seasonal structure of ionosonde Es parameters and meteoroid deposition rates. Planetary and Space Science, 32(12), 1533–1539. doi:10.1016/0032-0633(84)90021-7.

Kotadia, K. M., & Jani, K. G. (1967). Sporadic-E ionization and anomalous increase in the rate of radar meteor counts during 1963. Journal of Atmospheric and Terrestrial Physics, 29(2), 221–223. doi:10.1016/0021-9169(67)90137-7.

Reviewer: 8. Figure 5 should be in discussions instead of conclusions.

Authors' Response: Authors would like to thank reviewer for pointing this out. The manuscript has been thoroughly read by the authors and the language mistakes have been rectified, wherever possible.

Reviewer: 9. Lines 157-159: The authors could give more proposals because this behavior occurs over the analyzed region.

Authors' Response: Following sentences have been added considering reviewer's comments:

"This may have happened because plasma density abnormalities may exist which may cause ionograms to record scatter echoes beyond the foEs. The abnormalities are caused by plasma instabilities due to the various electrodynamic processes in the ionosphere. Meteoric activity may provide metallic ions to the ionosphere, but they may not be displayed in ionograms if the conditions are unfavorable. This may be why a good correlation between meteor activity and Es layer is not seen by our two collocated instruments. Such results have been rarely reported in the literature and do not comply with frequently reported studies which established a strong seasonal correlation between daily meteor counts with daily averages of Es layer occurrences, as mentioned in the references above."

Reviewer: 10. Do the authors only consider the nighttime values of foEs to perform the correlation in Figure 5? Does this anti-correlation in this Figure not occur because the Es layer is very dynamic?

Authors' Response: Both full day and nighttime foEs values are used to perform correlation analysis presented in Fig 5. We have presented both data sets. As shown in Fig 5, the red line (and dots) indicates the correlation pattern (individual relationships) of the temporal occurrences of foEs with respect to the meteor count. Since presence of Es layer is a daytime phenomenon, the relationship of the full day foEs averages and the meteor count would be considered as a legitimate comparison. However, pattern of nighttime foEs averages and the meteor count data (which is actually observed during nighttime) shows almost the similar relationship confirming the consistency of the results (blue line and dots in Fig 5).

Authors agree with the referee that the daily presence of the Es layer over Arabian Peninsula is very dynamic, however, the anti-correlation behavior presented in Fig 5 is not affected by this fact because the correlation behavior has been calculated using monthly averages which has smoothen out the daily fluctuations (as shown by thick blue/brown curves of monthly averages in Fig 4).

Reviewer: 11. There are many mistakes in the language (for example, relationship in Figure 5), and the English need to be improved.

Authors' Response: Authors would like to thank reviewer for pointing this out. The

manuscript has been thoroughly read by the authors and the language mistakes have been rectified, wherever possible.

Reviewer: 12. Please, consider including the word layer after the Es.

Authors' Response: The word layer has been included after Es in the revised manuscript (highlighted in red).

Please also note the supplement to this comment:
https://angeo.copernicus.org/preprints/angeo-2020-74/angeo-2020-74-AC1-supplement.pdf
* * *
0400 LT · 0500 LT · 0600 LT
0700 LT · 0800 LT · 0900 LT
1000 LT · 1100 LT · 1200 LT
1300 LT · 1400 LT · 1500 LT
1600 LT · 1700 LT · 1800 LT
1900 LT · 2000 LT · 2100 LT
2200 LT · 2300 LT · 0000 LT
0100 LT · 0200 LT · 0300 LT

**Fig. 1.**

---

## Author Comment (AC2) · 3 Jan 2021

Reviewer: The paper presents a topic of sure interest that can stimulate the curiosity of a number of scientists because it poses questions still unsolved and because the analysis is based on measurements taken in a region scarcely reported in the open literature. This is the reason why I would be in favour of the paper publication.

Authors' Response: Authors would like to thank reviewer for spending his/her precious time to read and comment on the paper. The comments from the reviewer have certainly helped improve the quality of the paper.

Reviewer: Nevertheless, my major concern is about the Es definition: are the authors considering the foEs or the appearance of Es? I see some confusion in the manuscript

between the two quantities and I wonder why the authors are accounting for foEs instead of focusing on the occurrence of Es.

Authors' Response: In this work, the foEs has been used as an indicator of the intensity of the presence of Es over the observed region. Authors had considered comparing individual events of Es with the presence of meteor counts but could not find a scientific way to do so since the meteor counts are done only during nighttime because of visual camera observations. At the other end, the Es is mostly present in the daytime hours and recorded through ionograms every 15 minutes. In order to compare the nighttime meteor count to the day/nighttime Es occurrences, it is understood that average daily Es intensity and daily meteor count would be appropriate measures to compare the trends of the number of meteors present and their impact on the presence of Es. This is not the first time that the comparison has been done in this way. Other authors (such as Haldoupis et al., 2007) have also used the same strategy to compare the two independent data sets for analysis. In order to make it clear for the reader, following sentence has been added at the end of 'Data and Methodology' section (highlighted in red in the revised manuscript):

"Since the data from meteor towers are only available from nighttime observations and the data from ionosonde is observed throughout the day and night, the daily Es intensity (average foEs value) has been used to compare with the daily meteor count to study the impact of the number of meteors present and their influence on the presence of Es (Haldoupis et al., 2007)."

Reviewer: Here follow some major issues that the authors might consider. In the Discussions: You write "A constant presence of Es can be observed throughout the year and all hours of the day", but the persistence throughout the year is not visible from your plot in which you identify the hourly average foEs as function of the day time. To show such consistency you should include the count.

Authors' Response: As per reviewer's request, the plot (attached as Fig. 1) shown
below has been included as Fig 3(a) in the revised manuscript to show consistent presence of Es occurrences (number of Es events) observed from the ionosonde; throughout the year as function of local time. Figure below clearly shows that a constant presence of Es occurrences can be observed on all hour of the day.

In order to be clear that the intensity of Es (average foEs values) shown in Fig 3(b) (Fig 3(a) in the original manuscript) is much higher around midday hours compared to early morning or evening hours, regardless of the number of Es occurrences. The sentence mentioned by the reviewer in the comment above has been rephrased as follows (highlighted in red in the revised manuscript):

"Fig 3(a) and 3(b) show that a constant presence of Es can be observed throughout the year and all hours of the day with higher intensity (average foEs) around midday hours and with lesser intensity at early morning and nighttime hours".

Reviewer: You write "The trend of monthly averages of Es clearly shows a rise in summer months and a decline in the autumn and winter months." From the plot I can see an increase in springtime and a decrease from early summer.

Authors' Response: The intention here was to convey that the trend of Es shows higher values in summer and lower values in winter months. Off course, the in-between transitions occur during in-between seasons of spring and autumn, respectively. However, authors agree with the reviewer that the statement is not stating the idea clearly and needs rephrasing. The rephrased sentence has been included in the revised manuscript as follows:

"The trend of monthly averages of Es layer intensity shows a maximum in late spring and early summer months and a minimum in winter months (except for a slight peak in January)".

Reviewer: In figure 5 are you reporting the foEs or the count of Es occurrence?

Authors' Response: Fig 5 is a relationship between Es layer intensity i.e. monthly
averages of foEs values. To clarify it, the caption of the figure has been modified with an indication that the Es intensity is actually monthly averages of foEs values (highlighted in red in the revised manuscript).

Reviewer: Minor revisions Fig.3 caption: meteor instead of metero Lines 106-110 page 4: replace o with âŮẹ Line 119 page 4: Fig 4 shows

Authors' Response: Corrected.

Please also note the supplement to this comment:
https://angeo.copernicus.org/preprints/angeo-2020-74/angeo-2020-74-AC2-supplement.pdf
* * *
**Fig. 1.**

---

## Referee Report (RR1)

angeo-2020-74    Submitted on 08 Nov 2020

Contribution of Meteor Flux in the Occurrence of Sporadic-E (Es) Layer over Arabian Peninsula

Muhammad Mubasshir Shaikh, Govardan Gopakumar, Aisha Abdulla Alowais, Maryam Essa Sharif, and Ilias Fernini

There are few papers dealing with the relationship between meteor events and Es occurrence and the region studied in the proposed manuscript is scarcely investigated. Thus the work can provide a valuable scientific contribution. Nevertheless, I find the scientific discussion too vague and I recommend to revise significantly the paper that cannot be accepted for publication in its present form.

My major concern is about the discussion that needs to be extended, integrated and tailored to the region under investigation.

I read the paper by Chandra et al. (2001) mentioned in the manuscript: the authors of that paper made an in-depth dissertation of the Es layer formation and evolution interpreting the shape of the traces in the ionograms and they provided a detailed description of the meteor event. I understand the authors consider several meteor events, nevertheless I invite them to extend their discussions. In particular, the discussion needs to be extended when the authors claim (lines 142-143) "The abnormalities are caused by plasma instabilities due to the various electrodynamic processes in the ionosphere." What are these various processes? Are these processes locally generated or are they linked to plasma transport? What do the authors mean with "unfavorable conditions" in the sentence (lines 143-145) "Meteoric activity may provide metallic ions to the ionosphere, but they may not be displayed in ionograms if the conditions are unfavorable."? Also the sentence (lines 153-155) "Es layer may be affected by differences in climatology and wind dynamics" should be extended and discussed in the context of the regional analysis presented in the manuscript.

Minor comments

Caption Figure 3: Simultaneous monitoring of meteors and Es layer over Arabian Peninsula from May 2019 – April 2020. (a) Es occurrence frequency as function of local time, (b) **Hourly average of foEs recorded using ionosonde**, (b) Hourly meteor count.

Figure3b Y-axis label: **foEs average (MHz)**

Line 116: **Fig 4 is a comparison between the daily and monthly meteor counts with daily and monthly averages of foEs** .

Caption Figure 4: Daily and monthly averages of **foEs** and meteor count over Sharjah. (a) Including all observations (24 hours), (b) Nightime observations only

Figure 4 Y-axis label: **foEs average (MHz)**

Caption Figure 5: Relationship betweeen **foEs**  monthly averages and monthly meteor count observed at Sharjah.

Figure 5 X-axis label: **foEs monthly average (MHz)**

---

## Referee Report (RR2)

angeo-2020-74    Submitted on 08 Nov 2020

Contribution of Meteor Flux in the Occurrence of Sporadic-E (Es) Layer over Arabian Peninsula

Muhammad Mubasshir Shaikh, Govardan Gopakumar, Aisha Abdulla Alowais, Maryam Essa Sharif, and Ilias Fernini

Dear Authors,

Your revised manuscript is now ready for publication after some minor revisions described below.

Minor comments

Lines 55-56: The dependence of Es layer intensity (foEs) due to meteor count  has been calculated using linear correlation coefficients.

Line 109: recorded as 69.43 **sfu**

Line 116: **Fig 4 shows a comparison between the daily and monthly meteor counts with daily and monthly averages of foEs**

Duplication of the text between lines 164-166 and lines 173-175. The same between lines 171-172 and lines 176-177. The remaining part of text from the line 173 to 177 has to be merged in the previous sentences.

---

## Author Response (AR4)

**Response to Reviewers' Comments**

All Responses

This paper is very interesting, but it is necessary significant improvements before publication. Please, consider the suggestions following:

**Authors' Response**
Authors would like to thank reviewer for spending his/her precious time to read and comment on the paper. The comments from the reviewer have certainly helped improve the quality of the paper. All changes made with reference to the reviewers' comments are highlighted in red in the revised manuscript.

**Reviewer 1**
1. The main point is the lack of manuscript organization. The authors need to include more information and the discussion about the anti-correlation.

**Authors' Response**
Considering reviewer's comment, following sentences have been added at the end of "Introduction" section:

"Correlation coefficient in this paper has been used to study the linear relationship of meteor counts and the intensity of Es layer in their diurnal and seasonal variation. On the contrary, anti-correlation would mean that variations of the two independent observations during the observational period have opposite linear trends."

**Reviewer 1**
2. Abstract- Lines 13-15: The authors need to include more details about the results.

**Authors' Response**
Last sentence of the Abstract has been modified as follows to indicate more details about the outcomes of the study:

"The trend of monthly averages of Es layer intensity shows a maximum in late spring and early summer months and a minimum in winter months whereas the meteor counts were highest in winter months and lowest in spring and early summer months. This shows that the presence of Es layer and the meteor counts have no correlation in time, both diurnally and seasonally, which leads us to conclude that the presence of meteors is not the main cause of the presence of Es layer over Arabian Peninsula."

**Reviewer 1**
3. Introduction: I suggest that authors include 2 or 3 paragraphs about the previous study of the Es layer behavior over the Arabian Peninsula and the wind shear and meter relationship.

**Authors' Response**
The authors were not able to find any previous studies about the relationship between windshear and meteors over the Arabian Peninsula. The only study reporting the observations of meteors over Arabian Peninsula is by Fernini et al., (2020)." Following para has been added for the previous studies found which studied the behavior of Es Layer:

"The behavior of Es layer over Arabian Peninsula has not been studied by many. Recently, Shaikh et al. (2020a; 2020b) demonstrated the relationship between L-band scintillation and the occurrence of the Es

layer over the Arabian Peninsula. The study also revealed a consistent presence of Es layer during the nighttime hours, between sunset and sunrise."

Shaikh, M., Gopakumar, G., Hussein, A., Kashcheyev, A., & Fernini, I. (2020b). Daytime GNSS scintillation due to Es over Arabian Peninsula during low solar activity. Results In Physics, 20, 103761. doi: 10.1016/j.rinp.2020.103761.

**Reviewer 1**
4. The last paragraph of the Introduction (lines 43-45) needs to be more specific with the authors' results in this work.

**Authors' Response**
Following line have been added at the end of the 'Introduction' section:

"It has been observed that the presence of meteors is not the main cause of the presence of nighttime Es over Arabian Peninsula since the Es layer intensity (average critical frequencies of the Es layers (foEs)) show no seasonal correlation with the number of meteors observed."

**Reviewer 1**

5. The authors need to explain Figure 1 and Figure 2 in more detail and maybe include the examples of ionograms since the Es layer are different forms in each region.

**Authors' Response**
Considering reviewer's comment, the first paragraph of the section 'Data and Methodology' has been modified to include more details about Figure 1, as follows:

"The meteor counts for this study has been obtained in collaboration with the UAE Meteor Monitoring Network (UAEMMN) project (Fernini et al., 2020). The project aims to monitor and detect meteors' occurrences in the region above the United Arab Emirates from sunset to sunrise. To do so, three monitoring towers have been constructed and installed in different parts of the country. For each tower, sixteen cameras are distributed along a ring-like structure with lenses of 6mm and 8mm, while the 17th camera utilizes a wide-angle lens and is located at the center of the structure (Fernini et al., 2020). Following a simulation using Systems Tool Kit software (STK: https://www.agi.com/products/50 stk) as shown in Fig 1a, the towers' locations were selected as illustrated in Fig 1b (made using © Google Maps). In Fig 1, green color represents the area of the sky covered by the 8mm lenses, while the red represents the coverage of the 6mm lenses. The yellow squares show what the wide-angle lens can see and cover. Thus, the STK simulation illustrates how much each tower covers of the UAE sky, and this adds up to 70% coverage of the sky. Each of the three UAEMMN towers employs the use of the UFOCapture Software developed by SonotaCo (SonotaCo, 2005) to detect meteor occurrences. The software can detect movements from the feed of the cameras on the towers. If a movement or action is detected, it writes the video of the action to the hard disk of the computer, from a few seconds before the action is recognized to a few seconds after the action is completed. During the night, the bright streaks produced by a meteor burning up in the atmosphere allows the software to easily detect movements from the sudden changes in pixel values."

The second paragraph of the section 'Data and Methodology' has been modified to include more details about Figure 2, as follows:

"Two other software, UFOAnalyzer and UFOOrbit, also developed by SonotaCo (SonotaCo, 2007a; SonotaCo, 2007b), are used to calculate parameters that define the meteorite. UFOAnalyzer can calculate the direction and elevation of the meteorite occurrence. If the meteorite is detected by two or more sites, UFOOrbit can calculate the orbit and the radiant point of the meteorite. Fig 2 shows a radiant map obtained as the result of analyses by the software. The radiant map shows radiant points on a sinusoidal

projection map of the observed meteors, which is defined as the point in the sky from which the path of the observed meteor begins. For a radiant point to be plotted on the map by the software, a double detection of the meteor should occur, meaning that two cameras from at least two different towers need to observe the same meteor. Fig 2 shows the radiant points of meteors observed by the Sharjah and Al-Yahar towers during the period between May 2019 and April 2020. On the map, constellations such Orionids and Taurids are denoted as J5_Orio, J5_nTa and sTa, respectively. Hence, the radiant points that are close to a constellation imply that they belong to the respective meteor group. In this figure, there are meteors that belong to the Orionids meteor shower, as well as Southern and Northern Taurids and several others, in addition to sporadic meteors that do not belong to any shower. By locating the radiant maps, the network ensures its accuracy in terms of linking a meteor to its respective shower.  The radiant velocity is color coded as shown in the figure."

Following figure shows selected ionograms obtained during a 24-hour period from LT 4:00 (UT 0:00) on July 31, 2019 to LT 4:00 on August 1, 2019. It can be observed that Es layer traces are present persistently over the 24-hour period. Most of the traces indicate an f-type (flat) Es layer, however, c-type Es layers are also seen infrequently.

[Figure]

**Reviewer 1**

6. Lines: 109-110: Haldoupis et al. (2007) only talk that the Es layers and the meteoric influx follow a similar seasonal dependence marked by a strong summer maximum only. The phrase seems that the results of Haldoupis et al. (2007) are related with both solstices.

**Authors' Response**

In the para mentioned by the reviewer (Lines: 109-110), authors have mentioned that there are numerous studies showing that the meteor count rate peaks in summer which was not observed by the data collected for this study. Haldoupis et al. (2007) showed the same by using radio frequency radar data from the mid-latitudes. In addition, in Fig. 2 of their article, Haldoupis et al. (2007) clearly showed that there is also a good correlation present between mean daily foEs and meteor counts throughout the year; in all four seasons.

**Reviewer 1**

7. The authors mentions that the "There are also numerous studies whose results are inconclusive", but they do not perform a deeply discussion about to works.

**Authors' Response**

Authors are thankful for the reviewer for pointing out the missing information. Following lines has been added in the manuscript for references:

"For example, Baggaley and Steel (1984) were not able to find any correlation between meteor activity and occurrence of Es layers. Kotadia and Jani (1967) reported that they did not find any increase in the occurrence of Es layers during a period of anomalously large increase in meteor incidence in 1963, but instead found that Es layers were formed less frequently during that period, suggesting an inverse relationship between the formation of Es layers or meteor incidents. The results presented in this paper also follow a similar pattern, with foEs decreasing significantly during the period between October 2019 to January 2020; even with the increased meteor count during that period (see Fig. 4)."

Baggaley, W. J., & Steel, D. I. (1984). The seasonal structure of ionosonde Es parameters and meteoroid deposition rates. Planetary and Space Science, 32(12), 1533–1539. doi:10.1016/0032-0633(84)90021-7.

Kotadia, K. M., & Jani, K. G. (1967). Sporadic-E ionization and anomalous increase in the rate of radar meteor counts during 1963. Journal of Atmospheric and Terrestrial Physics, 29(2), 221–223. doi:10.1016/0021-9169(67)90137-7.

**Reviewer 1**

8. Figure 5 should be in discussions instead of conclusions.

**Authors' Response**

Authors would like to thank reviewer for pointing this out. The manuscript has been thoroughly read by the authors and the language mistakes have been rectified, wherever possible.

**Reviewer 1**

9. Lines 157-159: The authors could give more proposals because this behavior occurs over the analyzed region.

**Authors' Response**

Following sentences have been added considering reviewer's comments:

"This may have happened because plasma density abnormalities may exist which may cause ionograms to record scatter echoes beyond the foEs. The abnormalities are caused by plasma instabilities due to the various electrodynamic processes in the ionosphere. Meteoric activity may provide metallic ions to the ionosphere, but they may not be displayed in ionograms if the conditions are unfavorable. This may be why a good correlation between meteor activity and Es layer is not seen by our two collocated

instruments. Such results have been rarely reported in the literature and do not comply with frequently reported studies which established a strong seasonal correlation between daily meteor counts with daily averages of Es layer occurrences, as mentioned in the references above."

**Reviewer 1**

10. Do the authors only consider the nighttime values of foEs to perform the correlation in Figure 5? Does this anti-correlation in this Figure not occur because the Es layer is very dynamic?

**Authors' Response**

Both full day and nighttime foEs values are used to perform correlation analysis presented in Fig 5. We have presented both data sets. As shown in Fig 5, the red line (and dots) indicates the correlation pattern (individual relationships) of the temporal occurrences of foEs with respect to the meteor count. Since presence of Es layer is a daytime phenomenon, the relationship of the full day foEs averages and the meteor count would be considered as a legitimate comparison. However, pattern of nighttime foEs averages and the meteor count data (which is actually observed during nighttime) shows almost the similar relationship confirming the consistency of the results (blue line and dots in Fig 5).

Authors agree with the referee that the daily presence of the Es layer over Arabian Peninsula is very dynamic, however, the anti-correlation behavior presented in Fig 5 is not affected by this fact because the correlation behavior has been calculated using monthly averages which has smoothen out the daily fluctuations (as shown by thick blue/brown curves of monthly averages in Fig 4).

**Reviewer 1**

11. There are many mistakes in the language (for example, relationship in Figure 5), and the English need to be improved.

**Authors' Response**

Authors would like to thank reviewer for pointing this out. The manuscript has been thoroughly read by the authors and the language mistakes have been rectified, wherever possible.

**Reviewer 1**

12. Please, consider including the word layer after the Es.

**Authors' Response**

The word layer has been included after Es in the revised manuscript (highlighted in red).

**Reviewer 1**

There are few papers dealing with the relationship between meteor events and Es occurrence and the region studied in the proposed manuscript is scarcely investigated. Thus the work can provide a valuable scientific contribution. Nevertheless, I find the scientific discussion too vague and I recommend to revise significantly the paper that cannot be accepted for publication in its present form.

My major concern is about the discussion that needs to be extended, integrated and tailored to the region under investigation. I read the paper by Chandra et al. (2001) mentioned in the manuscript: the authors of that paper made an in-depth dissertation of the Es layer formation and evolution interpreting the shape of the traces in the ionograms and they provided a detailed description of the meteor event. I understand the authors consider several meteor events, nevertheless I invite them to extend their discussions.

**Authors' Response**

Authors would like to thank reviewer for spending his/her precious time to read and comment on the paper. All comments and recommendations from reviewer to extend discussion have been addressed and a number of references have been added with reference to the added discussion. It is expected that the

extended discussion will satisfy the reviewer. The comments from the reviewer have certainly helped improve the quality of the paper. All changes made with reference to the reviewers' comments are highlighted in red in the revised manuscript.

**Reviewer 1**

In particular, the discussion needs to be extended when the authors claim (lines 142-143) "The abnormalities are caused by plasma instabilities due to the various electrodynamic processes in the ionosphere." What are these various processes? Are these processes locally generated or are they linked to plasma transport?

**Authors' Response**

To extend the discussion and incorporating the specific point raised by the reviewer, following sentences have been added in the revised manuscript:

"Cross-field plasma instabilities cause the abnormalities due to the various electrodynamic processes in the ionosphere. These instabilities are triggered by the enhancement of plasma density in a particular volume when an external force acts on that same volume. A small disturbance can then lead to the separation of charges, which produces a small electric field, which with the presence of the geomagnetic field increases the disturbance (Simon, 1963)."

Simon, A.: Instability of a Partially Ionized Plasma in Crossed Electric and Magnetic Fields. Physics of Fluids, 6(3), 382. doi:10.1063/1.1706743.

**Reviewer 1**

What do the authors mean with "unfavorable conditions" in the sentence (lines 143-145) "Meteoric activity may provide metallic ions to the ionosphere, but they may not be displayed in ionograms if the conditions are unfavorable."?

**Authors' Response**

To extend the discussion and incorporating the specific point raised by the reviewer, following sentences have been added in the revised manuscript:

"The aforementioned instabilities have been shown to be capable of producing the diffuse type of sporadic E (Tsuda et al., 1969). The formation of this diffuse layer may cause the ionogram to display scatter echoes that exceed the actual critical frequency of the sporadic E layer formed as a result of metallic ions deposited by meteors."

Tsuda, T., Sato, T., and Matsushita, S.: Ionospheric irregularities and the cross-field plasma instability. Journal of Geophysical Research, 74(11), 2923–2932. doi:10.1029/ja074i011p02923.

**Reviewer 1**

Also the sentence (lines 153-155) "Es layer may be affected by differences in climatology and wind dynamics" should be extended and discussed in the context of the regional analysis presented in the manuscript.

**Authors' Response**

To extend the discussion and incorporating the specific point raised by the reviewer, following sentences have been added in the revised manuscript:

"For example, long-period zonal and meridional winds at the mesopause region, with periods between 2 to 18 days may be considered to be planetary wave activity. Planetary waves have been observed to have strong variability between different seasons, with periods of 2 days in the summer, 5 days in spring and

even exceeding 10 days during the winter (Jacobi et al., 1998). Studies have proposed vortex flows associated with planetary waves to explain the season dependance of sporadic E layers (Shalimov et al., 1999). Vortex flows are already known to affect the development of E layers (Pancheva et al. 2003). The meteor count may also be influenced by some biases. A number of the recorded meteors may not be metallic in nature and would not deposit any metallic ions in the ionosphere, possibly explaining why a higher meteor count during winter months did not amount to a higher average foEs."

Pancheva D., Haldoupis C., Meek C. E., Manson A. H., and Mitchell N. J.: Evidence of a role for modulated atmospheric tides in the dependence of sporadic E on planetary waves. Journal of Geophysical Research, 108(A5), 1176, doi:10.1029/2002JA009788.

Shalimov S., Haldoupis C., Voiculescu M., and Schlegel K.: Midlatitude E region plasma accumulation driven by planetary wave horizontal wind shears. Journal of Geophysical Research, 104, 28,207.

Jacobi C., Achminder R., and Kürschhner D.: Planetary wave activity obtained from long-period (2– 18 days) variations of mesopause region winds over central Europe (52N, 15E). Journal of Atmospheic and Solar Terrestrial Physics, 60, 81.

**Reviewer 1**

Minor comments:

Caption Figure 3: Simultaneous monitoring of meteors and Es layer over Arabian Peninsula from May 2019– April 2020. (a) Es occurrence frequency as function of local time, (b) **Hourly average of foEs recorded using ionosonde**, (b) Hourly meteor count.

Figure3b Y-axis label: **foEs average (MHz)**

Line 116: **Fig 4 is a comparison between the daily and monthly meteor counts with daily and monthly averages of foEs** layer occurrences.

Caption Figure 4: Daily and monthly averages of **foEs** and meteor count over Sharjah. (a) Including all observations (24 hours), (b) Nightime observations only

Figure 4 Y-axis label: **foEs average (MHz)** Caption Figure 5: Relationship betweeen **foEs** layer monthly averages and monthly meteor count observed at Sharjah.

Figure 5 X-axis label: **foEs monthly average (MHz)**

**Authors' Response**
All the minor changes suggested by the reviewer have been incorporated in the revised manuscript (highlighted in red).

**Reviewer 1**

Dear Authors,
Your revised manuscript is now ready for publication after some minor revisions described below.

**Authors' Response**
Authors would like to thank the reviewers for their valuable comments that improved the quality of the paper significantly. By taking the advantage of this occasion, authors are also like to thank the topical editor for his/her guidance and support throughout the process from submission to acceptance.

Minor comments
Lines 55-56: The dependence of Es layer intensity (foEs) due to meteor count the has been calculated using linear correlation coefficients.

Line 109: recorded as 69.43 sfu

Line 116: Fig 4 shows a comparison between the daily and monthly meteor counts with daily and monthly averages of foEs

Duplication of the text between lines 164-166 and lines 173-175. The same between lines 171-172 and lines 176-177. The remaining part of text from the line 173 to 177 has to be merged in the previous sentences

**Authors' Response**
All the minor changes recommended by the reviewer have been incorporated in the revised manuscript including the removal of duplicate text.

**Reviewer 2**
The paper presents a topic of sure interest that can stimulate the curiosity of a number of scientists because it poses questions still unsolved and because the analysis is based on measurements taken in a region scarcely reported in the open literature. This is the reason why I would be in favour of the paper publication.

**Authors' Response**
Authors would like to thank reviewer for spending his/her precious time to read and comment on the paper. The comments from the reviewer have certainly helped improve the quality of the paper.

**Reviewer 2**
Nevertheless, my major concern is about the Es definition: are the authors considering the foEs or the appearance of Es? I see some confusion in the manuscript between the two quantities and I wonder why the authors are accounting for foEs instead of focusing on the occurrence of Es.

**Authors' Response**
In this work, the foEs has been used as an indicator of the intensity of the presence of Es over the observed region. Authors had considered comparing individual events of Es with the presence of meteor counts but could not find a scientific way to do so since the meteor counts are done only during nighttime because of visual camera observations. At the other end, the Es is mostly present in the daytime hours and recorded through ionograms every 15 minutes.

In order to compare the nighttime meteor count to the day/nighttime Es occurrences, it is understood that average daily Es intensity and daily meteor count would be appropriate measures to compare the trends of the number of meteors present and their impact on the presence of Es. This is not the first time that the comparison has been done in this way. Other authors (such as Haldoupis et al., 2007) have also used the same strategy to compare the two independent data sets for analysis.

In order to make it clear for the reader, following sentence has been added at the end of 'Data and Methodology' section (highlighted in red in the revised manuscript):

"Since the data from meteor towers are only available from nighttime observations and the data from ionosonde is observed throughout the day and night, the daily Es intensity (average foEs value) has been

used to compare with the daily meteor count to study the impact of the number of meteors present and their influence on the presence of Es (Haldoupis et al., 2007)."

**Reviewer 2**
Here follow some major issues that the authors might consider. In the Discussions: You write "A constant presence of Es can be observed throughout the year and all hours of the day", but the persistence throughout the year is not visible from your plot in which you identify the hourly average foEs as function of the day time. To show such consistency you should include the count.

**Authors' Response**

As per reviewer's request, the plot shown below has been included as Fig 3(a) in the revised manuscript to show consistent presence of Es occurrences (number of Es events) observed from the ionosonde; throughout the year as function of local time. Figure below clearly shows that a constant presence of Es occurrences can be observed on all hour of the day.

[Figure]

In order to be clear that the intensity of Es (average foEs values) shown in Fig 3(b) (Fig 3(a) in the original manuscript) is much higher around midday hours compared to early morning or evening hours, regardless of the number of Es occurrences. The sentence mentioned by the reviewer in the comment above has been rephrased as follows (highlighted in red in the revised manuscript):

"Fig 3(a) and 3(b) show that a constant presence of Es can be observed throughout the year and all hours of the day with higher intensity (average foEs) around midday hours and with lesser intensity at early morning and nighttime hours".

**Reviewer 2**
You write "The trend of monthly averages of Es clearly shows a rise in summer months and a decline in the autumn and winter months." From the plot I can see an increase in springtime and a decrease from early summer.

**Authors' Response**

The intention here was to convey that the trend of Es shows higher values in summer and lower values in winter months. Off course, the in-between transitions occur during in-between seasons of spring and autumn, respectively. However, authors agree with the reviewer that the statement is not stating the idea clearly and needs rephrasing. The rephrased sentence has been included in the revised manuscript as follows:

"The trend of monthly averages of Es layer intensity shows a maximum in late spring and early summer months and a minimum in winter months (except for a slight peak in January)".

**Reviewer 2**

In figure 5 are you reporting the foEs or the count of Es occurrence?

**Authors' Response**

Fig 5 is a relationship between Es layer intensity i.e. monthly averages of foEs values. To clarify it, the caption of the figure has been modified with an indication that the Es intensity is actually monthly averages of foEs values (highlighted in red in the revised manuscript).

**Reviewer 2**

Minor revisions Fig.3 caption: meteor instead of metero Lines 106-110 page 4: replace o with ∘ Line 119 page 4: Fig 4 shows

**Authors' Response**

Corrected.

**Reviewer 2**

This article had a significant improvement, and the results are significant. In this version, there are just too many small English mistakes. Some examples are listed below. However, I suggest that the authors revise these little mistakes in all the text.

**Authors' Response**

Authors would like to thank reviewer for reviewing their artcile. The comments from the reviewer have certainly helped improve the quality of the paper.

**Reviewer 2**

The article "the" is missing in a lot of parts. I cited here some examples, but I suggest the authors review the entire manuscript.
Line 10: Include the article "the" before Es layer.
Line 15-17: Please, divide the last sentence into two parts to avoid confusion to the reader. Also, include the article "the" before the Es layer.
Line 21: Include the article "the" before the word visible.
Line 22: Remove the comma after the word shower. And revise the comma in all the text.
Line 23: There are instead of there is.
Line 26: Include the article "the" before the word geographical.
Line 30: Include the article "the" before the word lower.
Line 31: Include the article "the" before the word radio.
Line 32: Replace this phrase with: "Detection using visual cameras can only be performed during the night compared to radio-based observations that can be performed throughout the day and suitable for estimating total meteor activity."
Line 36: Remove comma after Kopp.
Line 41: Include the word as after the as well.
Line 41/Line 42/Line 44/Line 46: Include the article "the" before Es layer.
Line 44: Include the article "the" before Arabian.
Line 49: Include the comma after the word paper. And the article "the" before Es layer in all the paragraphs.
Line 55-57: Rewrite the phrases.
Line 59: Replace for: "The meteor counts have been obtained".
Line 61: "To do so" is so informal. Please, replace it.

Line 62: Include the word with after the along.
Line 67: Replace for "Thus, the STK simulation illustrates how much each tower covers the UAE sky, which adds up to 70% coverage of the sky."

Line 73: Replace for "…to detect movements from the sudden changes easily in pixel values."
Line 95: "All the data used in this study are…" instead of "All the data used in this study is"
Line 105: Include the article "the" before Es layer.
Line 108: It is "a 1-year".
Line 109: It is "an average".
Line 109-110: Replace for "It is expected that the Es layer observations would be more substantial as the solar cycle gets stronger in the coming years."
Line 116: Replace for: "Fig 4 compares".
Line 117: Include the comma before the word "and" and remove it after the word nighttime.
Line 119: Replace for: "…the monthly meteor count shows an opposite trend with a larger number of meteors…"
Line 121: It is "a very similar…"
Line 122: It is "The meteor count is the same…"
Line 127: Include the comma after the Younger et al. (2009), and after Ascension Island (8oS).
Line 129: Include the article the before the word maximum.
Line 136: Replace for "Baggaley and Steel (1984) were unable to find any correlation between meteor activity and Es layers' occurrence."
Line 143: Replace for "Plasma instabilities cause the abnormalities due to the various electrodynamic processes in the ionosphere."
Line 146: Replace for "It is shown in Fig 5 that the annual variation of both observations, on average, does not correlate monthly, having linear correlation coefficients less than -0.35 (negative 0.35) for both full-day and nighttime observations."
Line 156-157: Replace for "unknown", include the article the before the word transportation, and remove the word also.
Line 174: Replace for "Es layer may not be observed if the meteoric activity period does not provide long-lived metallic ions in the background plasma density."
Line 176: Replace for "…, leading…"
Line 180: Replace for "of the role…"

**Authors' Response**
All changes suggested by the review have been incorporated in the revised manuscript and highlighted in red.